# Towards a Less Invasive Treatment for Head and Neck Cancer: Initial Evaluation of Gold Nanoparticle-Mediated Photothermal Therapy

**DOI:** 10.3390/pharmaceutics17101283

**Published:** 2025-10-01

**Authors:** Mariana Neves Amaral, Íris Neto, Mitza Cabral, Daniela Nunes, Elvira Fortunato, Rodrigo Martins, Carla Rodrigues, António P. de Almeida, José Catarino, Pedro Faísca, Hugo Alexandre Ferreira, João M. P. Coelho, Maria Manuela Gaspar, Catarina Pinto Reis

**Affiliations:** 1Research Institute for Medicines (iMed.ULisboa), Faculty of Pharmacy, Universidade de Lisboa, Av. Professor Gama Pinto, 1649-003 Lisboa, Portugal; marianaamaral@edu.ulisboa.pt (M.N.A.); irisneto@edu.ulisboa.pt (Í.N.); hugoferreira@campus.ul.pt (H.A.F.); jmcoelho@fc.ul.pt (J.M.P.C.); 2Instituto de Biofísica e Engenharia Biomédica (IBEB), Faculdade de Ciências, Universidade de Lisboa, Campo Grande, 1749-016 Lisboa, Portugal; 3CENIMAT|i3N, Department of Materials Science, School of Science and Technology, NOVA University Lisbon and CEMOP/UNINOVA, 2829-516 Caparica, Portugal; me.cabral@campus.fct.unl.pt (M.C.); daniela.gomes@fct.unl.pt (D.N.); emf@fct.unl.pt (E.F.); rfpm@fct.unl.pt (R.M.); 4REQUIMTE/CQFB, Departamento de Química, Faculdade de Ciências e Tecnologia, Universidade Nova de Lisboa, 2829-516 Caparica, Portugal; a3597@fct.unl.pt; 5Faculdade de Medicina Veterinária, Universidade de Lisboa, Av. Universidade Técnica, 1300-477 Lisboa, Portugal; alalmeida@fmv.ulisboa.pt; 6CECAV, Centro de Ciência Animal e Veterinária-Faculdade de Medicina Veterinária, Universidade Lusófona, Centro Universitário de Lisboa, Campo Grande 376, 1749-024 Lisbon, Portugal; p5663@ulusofona.pt (J.C.); pedrofaisca@ulusofona.pt (P.F.); 7I-MVET (Investigação em Medicina Veterinária), Faculdade de Medicina Veterinária, Universidade Lusófona, 1749-024 Lisbon, Portugal

**Keywords:** head and neck cancer, gold nanoparticles, photothermal therapy, novel therapies

## Abstract

**Background/Objectives:** Head and neck cancer (HNC) is the sixth most common cancer worldwide, with a high mortality, particularly from head and neck squamous cell carcinoma (HNSCC). Although some therapeutic strategies are available, they might cause severe side effects. For example, surgery may result in disfigurement and functional loss, severely impacting the patient’s quality of life. Thus, minimally invasive and more effective alternatives are needed. Gold nanoparticle (AuNP)-mediated photothermal therapy (PTT) is a promising approach for HNC, which relies on AuNP photothermal efficiency and tumor localization. This study aimed to synthesize and characterize AuNPs, evaluate their safety without laser activation, and assess their efficacy with laser activation. **Methods and Results:** Their physicochemical and photostability over three months and sterility were confirmed. In vitro safety was tested using human non-cancerous and HNC cell lines, while in vivo biocompatibility was evaluated in the hen’s egg chorioallantoic membrane (CAM) model, with no adverse effects observed. Upon laser activation, AuNPs reduced HNC cell viability by 50–70%, including HNSCC lines. In vivo biodistribution studies showed that AuNPs remained at the injection site for up to one month without toxicity. **Conclusions:** Overall, the developed AuNP formulation demonstrates stability, biocompatibility, and prolonged local retention, key attributes for effective and targeted PTT. These findings support the potential of AuNP-mediated photothermal therapy as a promising treatment modality for HNC, although further preclinical and clinical studies are needed to optimize treatment parameters.

## 1. Introduction

Head and neck cancer (HNC) is a group of malignancies that, as the name indicates, affects a variety of anatomical structures in the head and neck regions, including the lip, oral cavity, tongue, maxillary sinus, paranasal sinus, pharynx, larynx, salivary glands, thyroid, etc. [1,2]. Altogether, HNC represents the sixth most common cancer diagnosis [1]. The most prevalent subtype of HNC is head and neck squamous cell carcinoma (HNSCC), accounting for 90% of all HNC cases [1,3,4,5]. Moreover, HNSCC also accounts for the majority of HNC-related deaths, and the incidence of this subtype is projected to increase by 40% by 2040 [4]. The predicted rise in HNSCC incidence is related to the increased exposure to tobacco-driven carcinogens, heavy alcohol consumption, and infections by oncogenic forms of EBV and HPV [4,6,7]. Despite treatment advances, the five-year survival for patients diagnosed with HNSCC is less than 50% [8].

HNC in general, and particularly HNSCC, severely impacts both the anatomy and function of the affected structures. Moreover, the therapeutic strategies for HNC management, especially resection surgery, may lead to even more significant disfigurations and function limitations, brutally impacting the patient’s quality of life and appearance [5]. These severe cosmetic disfigurations can lead to a decline in the patient’s mental health [9]. Besides tumor resection surgery, HNC is usually treated with a combination of chemotherapy and radiotherapy [4,5,7,8]. Advancements in HNC treatment include immunotherapy that, although appearing to be very promising in clinical trials, does not show the same efficacy in clinical settings, i.e., immunotherapy did not show a benefit for HNC patient survival [10].

Thus, many challenges remain in the treatment of HNC including the disfigurement that often results from surgical intervention; the loss of function (e.g., speech and swallowing) and fibrosis that can occur after organ-preserving radiotherapy; and the toxicity and severe side effects that frequently occur after chemotherapy [11]. Altogether, the current approaches used in the treatment of HNC significantly impact the patient’s quality of life and consequently, their mental health, which is reflected in the fact that HNC patients have the highest rates of depression and suicide amongst all the types of cancer [5,9,10].

Photothermal therapy (PTT) aims to treat malignancies through the conversion of light into heat, causing localized hyperthermia that will induce cell death and consequently lead to tumor death [12]. This novel approach is minimally invasive and does not seem to cause significant systemic side effects [13]. To increase hyperthermia’s selectivity, PTT can be conjugated with photothermal agents (PTAs), such as gold nanoparticles (AuNPs). These nanosystems absorb the light radiation and convert it into heat, focusing the produced heat at the tumor site [14]. Besides AuNPs, there are many nano-based PTAs made from other materials such as copper selenide [15], carbon [16,17], tungsten [18,19], graphene oxide [16], black phosphorus [20], copper [21,22], etc. [23]. However, although these nanosystems possess very promising optical properties for PTA applications, they face several limitations in terms of low biocompatibility that could lead to toxicity [24,25,26,27,28]. Moreover, PTAs based on the mentioned materials are usually synthetized through complex processes and the resulting nano-PTAs present poor colloidal stability and undergo oxidative degradation [26,27,29]. On the other hand, AuNPs are chemically inert, biocompatible, and have an enhanced photothermal conversion ability due to the Localized Surface Plasmon Resonance Effect (LSPR). Upon irradiation with light with a specific wavelength (SPR band), the oscillating electromagnetic field interacts with the AuNPs’ conduction band electrons, causing them to undergo synchronized coherent oscillation, which consequently results in the absorption and scattering of the irradiated light [14,30,31,32,33,34,35,36,37,38,39]. Traditionally, different wavelengths of light can be used, such as microwave, visible, and ultraviolet (UV) light; however, for AuNP-mediated PTT, light at a wavelength that is able to penetrate the skin should be used [32]. Near-infrared (NIR) light (780–3000 nm according to ISO 20473) has this ability as its wavelength range coincides with the therapeutic window (650–950 nm); light with this wavelength range is less absorbed by water and by hemoglobin contained in the tissues and is thus able to better penetrate tissues [14,32,36].

HNC is a relatively superficial tumor as most of the anatomical structures it affects are not localized deep below the skin. Moreover, in cases where HNC is localized in deeper places, e.g., inside the nasopharynx or in the hard or soft palate, it can be assessed using a laser probe inserted through the mouth or nasal cavity. Previously, our group developed a novel formulation of AuNPs for the treatment of Anaplastic Thyroid Carcinoma (ATC), a very aggressive subtype of thyroid cancer (a form of HNC) [36]. Although this formulation presented promising results in vitro, the formulation was further optimized to maximize its photothermal conversion ability and consequently, maximize the produced hyperthermia upon laser irradiation [31]. This work aims to study the application of the optimized AuNPs for AuNP-mediated PTT of different subtypes of HNC, including HNSCC. For this, after synthesis, the physicochemical and optical properties of the AuNPs were characterized. The AuNPs’ stability and sterilizability were also studied. Furthermore, the efficacy of the AuNPs against different subtypes of HNC was assessed in vitro using several human and murine HNC cell lines. Moreover, the safety of this formulation was assessed in in vitro, in ovo, and in vivo models. The biodistribution of AuNPs was assessed over 30 days after their administration.

## 2. Materials and Methods

### 2.1. Materials

Gold (III) chloride trihydrate (HAuCl_4_·3H_2_O), L-ascorbic acid, silver nitrate (AgNO_3_), rosmarinic acid, and 3-(4,5-dimethylthiazol-2-yl)-2,5-diphenyltetrazolium bromide (MTT) were acquired from Sigma-Aldrich (St. Louis, MO, USA). Roswell Park Memorial Institute (RPMI) 1640 medium with HEPES, Dulbecco’s modified Eagle’s medium (DMEM), fetal bovine serum (FBS), L-glutamine, penicillin, and streptomycin were supplied by Gibco (Thermo Fisher Scientific, Waltham, MA, USA). Fluid Thioglycolate Medium and Soybean Casein Digest Medium were supplied by Scharlau (Scharlab, Barcelona, Spain). Nitric acid (HNO_3_) Suprapur^®^ for Trace Analysis (69%) and hydrochloric acid (HCl) Suprapur^®^ for Trace Analysis (37%) were supplied by Carlo Erba (CARLO ERBA Reagents, Milan, Italy). Water Milli-Q was provided by a Merck Millipore System (Burlington, MA, USA). All other chemicals were of analytical grade.

### 2.2. Instruments

The following instruments were used: an Heidolph MR3001 magnetic stirrer hotplate (Heidolph Instruments, Schwabach, Germany), Z 327 K centrifuge (Hermle LaborTechnik GmbH, Wehingen, Germany), Ultima Spectrometer (Horiba Jobin-Yvon, Longjumeau, France), Zetasizer Nano S (Malvern Instruments, Malvern, UK), Zetasizer Nano Z (Malvern Instruments, Malvern, UK), Hitachi Regulus 8220 Scanning Electron Microscope (Hitachi, Mito, Japan), Shimadzu UV-1280 (Shimadzu, Corp., Kyoto, Japan), Microfil™ filtration system (Merck Millipore, Burlington, MA, USA), BioTek^TM^ EL x800^TM^ Absorbance Microplate Reader (Fisher Scientific, Hampton, NH, USA), Olympus CX51 microscope (Olympus Corporation, Tokyo, Japan), NanoZoomer-SQ Digital slice scanner C13140-01 (Hamamtsu, Japan), and Toshiba Astelion 16 CT scanner (Toshiba, Tokyo, Japan).

### 2.3. Cell Lines and Cell Culture

C643 and C3948 were kindly provided by Prof. Branca M. Cavaco (UIPM, IPOLFG) and grown according to their instructions. SALTO cells were kindly provided by Prof. Federica Cavallo (University of Torino) and Prof. Pier Luigi Lollini (University of Bologna) and grown according to their instructions. The other cell lines (RPMI 2650, Detroit-562 and HaCat) were grown according to the supplier’s instructions (ATCC^®^). 

The human ATC BRAF^wt^ cell lines C643 (commercial cell line) and C3948 were maintained in RPMI 1640 with HEPES supplemented with 10% FBS, 1% L-glutamine, 100 IU/mL of penicillin, and 100 µg/mL of streptomycin. C3948 was established from the fine-needle aspiration cytology of an unresectable human ATC [40].

The commercialized human cell lines RPMI 2650 (nasal HNSCC; ATCC^®^ CCL-30^TM^) and Detroit-562 (pharyngeal HNSCC; ATCC^®^ CCL-138^TM^) were maintained in RPMI 1640 and DMEM, respectively, supplemented with 10% FBS, 100 IU/mL of penicillin, and 100 µg/mL of streptomycin.

The mouse Neu-overexpressing salivary cancer cell line SALTO was maintained in Dulbecco’s modified Eagle’s medium (DMEM) supplemented with 20% FBS, 100 IU/mL of penicillin, and 100 µg/mL of streptomycin. These cells were established from mice salivary carcinoma arising in BALB-*neu*T transgenic male mice hemizygous for the p53^172R-H^ transgene driven by the whey acidic protein promoter [41].

The commercialized ATC human keratinocyte cell line HaCat was maintained in DMEM supplemented with 10% FBS, 100 IU/mL of penicillin, and 100 µg/mL of streptomycin.

All cell lines were kept in a humidified chamber at a temperature of 37 °C in an atmosphere of 5% CO_2_. Cell cultures were maintained every 2–3 days, and the medium was changed when a confluence of 80% was reached.

### 2.4. Animals

Female BALB/c mice were purchased from Charles River Laboratories (Barcelona, Spain). The animal housing was kept at a controlled temperature of 22.0 ± 4.0 °C, humidity of 50 to 65%, and in a 12 h light/dark cycle in the Faculty of Pharmacy’s facilities. The animals were kept under standard hygiene conditions, fed commercial chow, and given acidified drinking water ad libitum.

All experiments using animals were conducted in accordance with the guidelines of the animal welfare organ (ORBEA) of the Faculty of Pharmacy, Universidade de Lisboa; the EU Directive 2010/63/UE; and Portuguese laws (DL 113/2013, 2880/2015, 260/2016, and 1/2019) for the use and care of animals in research. The study was conducted under the protocol approved by the ORBEA with code number 01/2023 (communication of the approval on 29/03/2023). The experiments were approved by the national authority *Direção Geral de Alimentação e Veterinária* (DGAV).

### 2.5. Methods

#### 2.5.1. Preparation of AuNPs

The AuNPs were prepared following a previously described method [31]. Briefly, aqueous solutions of HAuCl_4_·3H_2_O, AgNO_3_, rosmarinic acid, and L-ascorbic acid were prepared. The solutions were then added to a beaker, under magnetic stirring (800 rpm), in the following order: aqueous solution of HAuCl_4_·3H_2_O; aqueous solution of L-ascorbic acid; aqueous solution of AgNO_3_; and aqueous solution of rosmarinic acid. After 15 min, the synthesis was complete and the resulting AuNP-containing suspension was stored for 24 h at 4 °C. After this, the AuNPs were recovered by centrifuging the suspension at 1520× *g* for 20 min using a Z327 K centrifuge. The supernatant was discarded and the AuNP-containing pellet was suspended in Milli-Q water [31].

The concentration of gold (Au) in the resulting AuNPs was determined by Inductively Coupled Plasma–Optical Emission Spectroscopy (ICP-OES). Prior to ICP-OES analysis, nitric acid (HNO_3_) was used to digest the AuNPs in the suspension [31].

#### 2.5.2. Physicochemical Characterization

##### Mean Size, Polydispersity Index (PdI), and Surface Charge

The mean size, polydispersity index (PdI) and surface charge of the AuNPs were determined. The AuNPs were diluted in Milli-Q water (1:10 v/v AuNPs/Milli-Q water) to determine mean size and polydispersity index (PdI) using Dynamic Light Scattering (DLS), and in PBS (1:10 v/v AuNPs/PBS) to determine the zeta potential using the Electrophoretic Mobility Assay. All measurements were performed ≥3 times.

##### Morphology Assessment

The shape and surface morphology of the AuNPs was assessed by Scanning Electron Microscopy (SEM). For this, 20 µL of an aqueous solution of the AuNPs (2500 µM gold) was placed on a silicon substrate and dried at room temperature (RT). SEM images were acquired using the mentioned microscope, and the images were digitally recorded.

##### Absorbance Spectrum Determination

The wavelength of the maximum absorbance peak (Abs_max_) and absorbance at 808 nm (Abs_808_) of the produced AuNPs were determined by scanning the absorbance spectrum (400–1000 nm) using a UV–Vis spectrophotometer. The determination of the absorbance spectra was performed >3 times.

#### 2.5.3. Stability Assessment over 3 Months

After preparation, the AuNP aqueous solution was stored at 4 °C for three months in order to assess the AuNPs’ stability when kept in water. For this, the different physicochemical properties that were previously mentioned (i.e., mean size, PdI, surface charge, absorbance at 808 nm, and morphology) were determined at different time points over the three months: day 0 (day of preparation), and 1, 2, and 3 months.

#### 2.5.4. Sterility Assessment

The AuNPs were prepared under sterile conditions. For this, all the prepared solutions used in the synthesis of the AuNPs were sterilized by filtration using syringe filters with pore size of 0.22 µm, and the AuNPs were prepared following the method described in Section 2.5.1. under aseptic conditions within a laminar flow chamber using previously autoclaved glassware. The AuNPs were then stored accordingly.

The sterility of the resulting AuNPs was assessed using two methods following the Eur. Pharmacopeia (Ph. Eur. 10th Edition): direct inoculation and membrane filtration. In both methods, Fluid Thioglycolate Medium (FTM) and Soybean Casein Digest Medium (TSB) were prepared according to manufacturer’s instructions for the detection of anaerobic and some aerobic bacteria (incubated at 30–35 °C) and the detection of fungi and aerobic bacteria (incubated at 20–25 °C), respectively.

##### Sterility Assessment by Direct Inoculation

The culture media (FTM and TSB) were directly seeded with the produced AuNPs (<10% of medium volume) and left to incubate at the indicated temperatures (30–35 °C for FTM and 20–25 °C for TSB) for 14 days. During this incubation period, the cultures resulting from the direct inoculation were visually inspected after the 14 days to detect the presence of any microbiological growth. The assessment was performed in quintuplicate.

##### Sterility Assessment by Membrane Filtration

The prepared AuNPs were filtered through membranes with a pore size of 0.45 mm using a Microfil™ filtration system. After that, the entire filtered sample was transferred to culture medium (FTM and TSB) and left to incubate at the indicated temperatures (30–35 °C for FTM and 20–25 °C for TSB) for 14 days. As with the previous assessment, the cultures resulting from the direct inoculation were visually inspected several times during the incubation period to detect the presence of any microbiological growth. The assessment was performed in quintuplicate.

#### 2.5.5. In Vitro Safety Assessment in Immortalized Cell Lines

Different human HNC cell lines (ATC, C643, and C3948), a murine HNC cell line (salivary HNSCC SALTO), and a non-cancer human cell line (HaCat keratinocytes) were used to assess the safety of the developed AuNPs that have not been activated by a laser. For this, different concentrations of AuNPs (125, 250, and 375 µM gold) were used. Each different cell line was seeded the day before incubation with the formulations at a concentration of 5 × 10^4^ cells/mL in 96-well plates and were left overnight to adhere in a controlled atmosphere of 5% CO_2_ at 37 °C. The following day, the cell medium was removed and replaced with AuNPs at the concentrations mentioned above or with complete medium, which served as the negative control (100% cell viability). The cells were left to incubate for 24 h. Afterwards, cell viability was assessed using the 3-(4,5-dimethylthiazol-2-yl)-2,5-diphenyltetrazolium bromide (MTT) assay [31,33,34,35,36,38,42]. The medium was removed from the wells, and the cells were washed twice with PBS (pH 7.4, USP32). After that, 50 µL of MTT (Sigma-Aldrich, St. Louis, MO, USA) at a concentration of 0.5 mg/mL (in incomplete medium) was added to the wells and incubated (at 37 °C with 5% CO_2_) until formazan crystals formed (~2–3 h). Finally, the crystals were solubilized using dimethyl sulfoxide (DMSO), and the absorbance was measured at 570 nm using a microplate reader. As previously mentioned, the absorbance of the control cells incubated with complete medium represented 100% cell viability. The results were calculated using Equation (1):(1)$$Cell\;Viability\;\left(\%\right)=\;\frac{{Abs}_{test\;group}}{{Abs}_{control}}\;\times \;100$$

#### 2.5.6. Efficacy in HNC Cell Lines 

HNC cancer cell lines, both human (RPMI 2650, Detroit 562, C643, and C3948) and murine (SALTO), and the non-cancer cell line HaCat were used to study the efficacy of the developed AuNPs at a concentration of gold of 250 µM upon laser activation irradiation. This allowed us to preliminarily study the efficacy of the AuNPs against HNC in vitro. Like before, each cell line was seeded at a concentration of 5 × 10^4^ cells/mL in 96-well plates and were left to adhere overnight (at 37 °C with 5% CO_2_). The layout of the plate was designed to ensure that the wells containing cells to be irradiated were separated from each other in all directions by one empty well. This design aimed to avoid collateral scattering and reflection of light from irradiated wells and to avoid secondary activation of the AuNPs and ensure that the AuNPs are only activated upon irradiation of their well. Like in the in vitro safety assessment, the cell medium was removed the next day and the cells were incubated with the AuNPs at a concentration of 250 µM of gold (test wells) or complete medium for 4 h. Subsequently, medium was removed and replaced with fresh complete medium and the cells were irradiated with a laser emitting 808 nm light (5.75 W/cm^2^, 5 min) using an RLTMDL-808-5W-5 diode laser coupled to an optical fiber with a 0.22 numerical aperture (Roithner LaserTechnik GmbH, Vienna, Austria) [31,33,35,36,38]. As for the in vitro safety assessment, 24 h after the laser irradiation, viability was assessed using the MTT assay. The medium was removed and the cells were washed twice with PBS (pH 7.4). Afterwards, 50 µL of MTT reagent (0.5 mg/mL) in incomplete DMEM was added to all the wells. After formazan crystals formed, they were solubilized using DMSO and the absorbances were read at 570 nm using a microplate reader. The absorbance of untreated cells represented 100% cell viability. The results were calculated using Equation (1).

#### 2.5.7. Hen’s Egg Test on the Chorioallantoic Membrane

Fertilized hen’s eggs were incubated at 37 °C and 60% humidity and rotated every 8 h. On day 9 post-fertilization, a small window was opened to access the CAM [43]. The eggs were then grouped as follows: an AuNPs-only group in which the eggs received AuNPs at a final concentration of 250 µM of gold; a laser-only group where the eggs were irradiated without receiving any other treatment; a AuNPs + Laser group in which the eggs received AuNPs at a final concentration of 250 µM of gold, followed by irradiation with a laser; a positive control group where NaOH (0.1 M) was added to the eggs; and a negative control group given PBS (pH 7.4). For the laser irradiation, the previously described laser was used. All tests were performed in triplicate. Visual inspection of the egg’s CAM was carried out at different time points (0, 1, 3, and 5 min) during the incubation of the formulations with the CAM (AuNP groups and positive and negative control groups). At the same time points during laser irradiation (AuNPs + Laser and laser groups), the CAM was checked for hemorrhaging, vascular lysis, or clotting.

#### 2.5.8. In Vivo Biodistribution and Safety Assessment

The animals were randomly distributed into three groups: animals that only received AuNPs subcutaneously (s.c.) (*n* = 4), animals that only received laser irradiation (0.59 W/cm^2^, 25 s, *n* = 3), and animals that received AuNPs s.c., which were activated 4 h post-administration by laser irradiation (0.59 W/cm^2^, 25 s; *n* = 4). The AuNPs were administered at a dose of 3.1 mg/kg of body weight. The day after the AuNP administration and laser irradiation (if applicable), the animals in each group were anesthetized and sacrificed by cervical dislocation following animal welfare principles and recommendations. After this, the main metabolism and excretion organs (i.e., liver, kidneys, and spleen) of each animal, as well as the site of injection and/or laser irradiation, were collected for histological analysis, determination of the elemental gold amount through inductively-coupled plasma optical emission spectroscopy (ICP-OES) analysis, and determination of tissue indexes (all animals in the groups) [38].

The gold content of the main metabolism and excretion organs (i.e., kidney, liver, and spleen), as well as whole blood, of the different groups was determined by ICP-OES. For this, the samples were frozen, freeze-dried, and micronized and homogenized through crushing. The spleen and blood samples, as well as parts of the homogenized liver and kidney samples, were weighed and subjected to microwave digestion. For the digestion, the sample and a mixture of nitric (HNO_3_) and hydrochloric (HCl) acids (3:1 v/v HCl:HNO_3_) were added to the digestion vessel, and two heating ramps were used: 120 °C for 5 min and 180 °C for 5 min. The resulting solutions were diluted to a final volume of 5 mL with Milli-Q water and the gold content was quantified [38].

To determine the tissue indexes of the main metabolism and excretion organs of the different animals, the organs were harvested and weighed, and the following equation was used (Equation (2)):(2)$$Tissue\;Index=\;\sqrt{\frac{Organ\;Weight}{Animal\;Weight}}\times 100$$

For the histological analysis, the tissues were fixed in 10% formalin and processed for standard Hematoxylin and Eosin (H&E) staining. An Olympus CX51 microscope was used to perform the histopathological assessment. A C13140-01 slice scanner was used to perform whole-slide scanning and representative photos were taken using the NDP.View2 Software v2.9.29 (Hamamatsu Photonics K.K., Shizuoka, Japan).

#### 2.5.9. CT Scan to Measure Local Retention of AuNPs in Vivo

Besides the study groups, the mice were grouped and subcutaneously administered AuNPs (3.1 mg/kg of body weight). Prior to the administration of AuNPs, the animals were anesthetized. Under anesthesia, the animals underwent computed tomography (CT) analysis weekly for up to 1 month. Images were acquired using a Toshiba Astelion 16 CT scanner (Minato, Tokyo, Japan) and the following parameters: 120 kV; 250 mA; rotation time of 0.75 s; slice thickness of 0.5 mm; pitch factor of 0.688; and 11 helical pitches with 16 rows. Horos (version 3.3.6, HorosProject sponsored by Nimble Co LLC d/b/a Purview, Annapolis, MD, USA), a DICOM image processing software, was used for the image analysis.

#### 2.5.10. Statistical Analysis

All data are expressed as the mean ± standard deviation (SD) and the sample size was *n* ≥ 3. Student’s *t*-test was used to compare two groups while one-way or two-way ANOVA was used to compare three or more groups. GraphPad Prism 10^®^ (GraphPad Software, San Diego, CA, USA) was used to carry out all the statistical analyses, and differences with a *p* < 0.05 were deemed significant.

## 3. Results and Discussion

### 3.1. Physicochemical Characterization

AuNPs were synthesized following a previously optimized method [18], and the resulting AuNPs were characterized in terms of their main physicochemical properties: mean size and size distribution (polydispersity index (PdI)), surface charge, and absorbance at 808 nm. The results are presented in Table 1.

The AuNPs presented a mean size of 118.2 ± 7.1 nm, with a PdI of 0.27 ± 0.05, indicating a homogenous size distribution. These particles are intended to be administered in situ, directly into the tumor, and remain retained at the administration site where they will exert their photothermal action upon laser activation. Thus, the AuNPs should ideally have a size between 100 and 200 nm as this is the size range that, according to the literature, remains retained in tumors [44,45]. The size distribution of the colloidal suspensions is a very important parameter in order to predict the optical and biological behaviors of AuNPs [34,46]. If the AuNP dispersion presents an homogenous size distribution (PdI < 0.4), then all the AuNPs will behave in similar ways [34,47]. The prepared AuNPs met both criteria: the mean size was between 100 and 200 nm (~118 nm) and they had a PdI < 0.40 (~0.27).

Regarding their surface charge, the AuNPs presented a surface charge of −32.9 ± 0.4 mV. Surface charge has a significant impact on NP behavior, and NPs can have a positive (>30 mV), neutral (between 10 and −10 mV), or negative (<−30 mV) charge [48,49]. Neutral AuNPs tend to be unstable and thus can coagulate, agglomerate, or flocculate when stored. Strongly charged particles (positive or negative) tend to show better stability due to stronger electrostatic repulsion [48]. Negatively charged AuNPs with zeta potential values below −30 mV are considered physically stable due to a sufficient electrostatic repulsion, preventing nanoparticle aggregation [50,51]. The AuNPs developed in this study exhibited negative surface charges. Moreover, and as demonstrated in previous work, the developed AuNPs only exhibited gold in their composition, thus evidencing a non-reactive surface resulting from the inert nature of this metal [31]. Together, these features are expected to provide long-term colloidal stability through electrostatic stabilization and the intrinsic chemical inertness of gold. Therefore, since AuNPs have a negative surface charge, they are expected to remain physically stable during long-term storage. This stability results from electrostatic repulsion between the suspended AuNPs as well as their surface chemistry, as demonstrated in this work after 3 months of storage.

As the laser used to activate the AuNPs emits at 808 nm, the absorbance of the AuNPs at this wavelength was determined. The prepared AuNPs presented an absorbance at 808 nm of 0.342 ± 0.007 a.u. at a concentration of 250 µM of gold. The prepared AuNPs showed a high molar extinction coefficient at 808 nm (~6 × 10^8^ M^−1^ cm^−1^), which translated into a high photothermal conversion ability, with the AuNPs (at 125 µM of gold) showing a temperature increase of ~10 °C upon laser irradiation (808 nm, 5.6 W/cm^2^) [31]. Moreover, the surface morphology of the AuNPs was assessed by SEM, and representative images can be seen in Figure 1. It should be noted that the AuNPs presented a compact nanoflower-like morphology with branches, as was previously described [31], representing an almost quasi-spherical morphology. This morphology seems to originate from the controlled agglomeration of gold nanosheets [31,52,53]. Nanoflowers show enhanced photothermal conversion due to the presence of branches as these structures act as “hot spots” [53,54].

### 3.2. Stability Assessment over 3 Months

The stability of the AuNPs over a period of 3 months, when kept in water and stored at 4 °C, was assessed by determining their mean size, PdI, surface charge, absorbance at 808 nm, and morphology. The results of these assessments after 0 (at time of synthesis), 1, 2, and 3 months are presented in Figure 2. The AuNPs’ mean sizes remained stable over the assessment period, varying by less than 10%. The mean size varied by less than 1% during the stability assessment period. The PDI value remained similar (below 0.3) over the analyzed period. The stable mean size and PdI values were corroborated by the morphology (SEM) analysis. Regarding the surface charge, it was possible to see that after 3 months, the zeta potential remained strongly negative (below −30 mV). However, the zeta potential was −32 mV (t = 0) and shifted to −43 mV at 2 months and −41 mV at 3 months post-synthesis. Since the AuNPs were not coated and were stored in Milli-Q water, the observed slight increase in negative surface charge is likely due to gradual adsorption of hydroxyl and other anionic species that were present in the water. Importantly, the mean size and PdI remained stable over the analyzed period (Figure 2A,B), indicating that the nanoparticles maintained colloidal stability despite the slight variation in zeta potential. Concerning the absorption of the AuNPs at 808 nm, it remained stable throughout the analysis period. Overall, it is possible to conclude that the AuNPs’ physicochemical properties remained stable when stored in water at 4 °C.

### 3.3. Sterility Assessment

In order for AuNPs to be used for health and treatment applications, their sterility needs to be ensured [55,56]. It is known that sterilization procedures like heat can harm AuNPs [57]. To ensure that AuNPs can be produced under sterile conditions, different batches of AuNPs were synthesized in a laminar flow chamber using the same four solutions after the initial solutions of the different components were sterilized by filtration (pore size of 0.22 µM). After AuNP preparation, their sterility was tested using two methods: direct inoculation and membrane filtration. In both assessments, the AuNPs were deemed sterile as no signs of microbial growth (e.g., medium clouding) were seen upon macroscopical inspection.

### 3.4. In Vitro Safety Assessment

When not irradiated by the laser, and thus in their inactive state, the AuNPs must be harmless to tissues and considered safe. Thus, the safety of the inactive AuNPs was preliminarily assessed in vitro following its incubation with human keratinocytes (HaCat) and human HNC (C643 and C3948, both ATC) and murine HNSCC (SALTO) cell lines. All the cell lines were incubated with the AuNPs at different concentrations of gold (125, 250, and 375 µM) for 24 h. Figure 3 presents the results of the in vitro safety assessment. It should be noted that the AuNPs are considered safe in vitro at the tested conditions as the viability of all the cell lines was greater than 70% (red dashed line). Due to this finding and based on previous work [31], the intermediate concentration of 250 µM of gold was selected for the preliminary in vitro efficacy assessment.

### 3.5. In Vitro Efficacy

The AuNPs’ efficacy was preliminarily assessed in vitro using different non-cancer and cancer cell lines: human keratinocyte (HaCat), human ATC (C643 and C3948), human HNSCC (Detroit-562 and RPMI 2650), and murine HNSCC (SALTO) cell lines. As mentioned above, the concentration of 250 µM of gold was selected for the present assessment. The cell lines were incubated with the AuNPs at a concentration of 250 µM of gold for 4 h to allow for internalization of the AuNPs by the selected cells. After this time, the AuNP-containing cell medium was replaced with fresh medium, and the cells were activated by 808 nm laser irradiation for 5 min at an irradiance of 5.75 W/cm^2^. The results are presented in Figure 4. The AuNPs at a gold concentration of 250 µM, when activated by a laser, led to a significant reduction in the cell viability of all HNC cell lines (ranging from 24.7 to 51.2%). However, in the non-cancer cell line HaCat, the decrease in cell viability was much less pronounced (less than 30%). In the HNC cell lines, activated AuNPs at a concentration of 250 µM of gold reduced cell viability to 24.9% ± 4.0% in C643; 24.7% ± 2.6% in C3948; 51.2% ± 9.1% in SALTO; 28.3% ± 5.9% in Detroit-562; and 28.4% ± 5.3% in RPMI 2650. Taking these results into account, the tested AuNPs, upon laser activation, were found to be very effective in reducing the viability of HNC cells without reducing HaCat cell viability below the defined cytotoxicity threshold of 70%.

### 3.6. Hen’s Egg Test on the Chorioallantoic Membrane

The chorioallantoic membranes of hen eggs were used to determine the irritative potential of the AuNPs following the Interagency Coordinating Committee on the Validation of Alternative Methods (ICCVAM) guidelines. The popularity of this in ovo toxicological assay is growing rapidly as it is an alternative to in vivo screening using rodents [58]. Thus, this model can be considered as a bridge between in vitro and in vivo testing, and can be used for an initial screening of the biocompatibility of AuNPs; if AuNPs are deemed toxic in the HET-CAM, there is no need to test them in murine models [59,60]. The goal of this assessment was to ensure that the AuNPs and the laser alone cannot irritate the CAM. However, when combined, the irradiated AuNPs produce heat. Thus, when AuNPs are activated by laser irradiation, heat is expected to be produced and consequently, the CAM might show some signs of irritation (e.g., hemorrhaging, vascular lysis, or clotting). However, to translate AuNP-mediated PTT into the clinic setting, we need to ensure that although heat is produced, this heat is not harmful to the patient and is considered safe.

The irritative potential of the different groups were tested on the CAM (negative control, positive control, AuNPs, laser, and AuNPs and laser combined). AuNPs were used at a concentration of 250 µM of gold and the laser, emitting at 808 nm, delivered an irradiance of 5.75 W/cm^2^. Although the CAM was exposed to the different treatments (AuNPs and laser alone and AuNPs and laser combined) for 5 min, images were recorded at different time points during this exposure (0, 1, 3, and 5 min). Representative images of the CAM exposed to the different treatments at different time points are shown in Figure 5. It is possible to see that exposure to the AuNPs or laser alone did not result in CAM irritation at the tested time points. However, the representative pictures at the different time points of the exposure of the CAMs to laser-irradiated AuNPs (AuNPs 250 µM + Laser) show notable CAM irritation starting from the 3 min time point. Due to the presence of laser irradiation, the CAM was harder to see and examine. Therefore, Figure 6 was added to show a clear representative picture of the CAM after 5 min of exposure to laser-irradiated AuNPs (+AuNPs) or to the laser alone (−AuNPs). By examining the two images in this figure, it is very noticeable that after 5 min of exposure to AuNPs irradiated by the laser, the CAM was irritated, as demonstrated by the increased number of blood vessels, as well as the presence of vascular lysis and hemorrhages. By examining the picture of the CAM that was exposed to laser-irradiated AuNPs (+AuNPs) for 5 min, it is possible to see a visibly burned CAM. This burn can also be observed in Figure 5 in the CAM exposed to the same treatment (AuNPs 250 µM + Laser), which appears as white dots in the irradiated area. Thus, the increase in these white dots over time indicated that the CAM is becoming more damaged over time, starting at around 2 min, which is clearly visible at the 3- and 5-min time points.

### 3.7. Acute Biodistribution and Safety

The safety and acute biodistribution of the AuNPs were assessed in wild-type BALB/c female mice. The animals were divided into three treatment groups: laser, AuNP, and laser-activated AuNP groups. For this assessment, the animals in the AuNP and laser-activated AuNP groups received a single subcutaneous administration of the AuNPs (3.1 mg/kg of body weight) in the neck region. After 4 h, the animals in the laser and laser-activated AuNP groups were irradiated for 25 s with an irradiance of 5.75 W/cm^2^. All the animals were sacrificed the following day and the tissue indexes of the main excretion organs (kidneys, spleen, and liver) were determined. The tissue indexes are presented in Table 2. It was found that there were no differences among the three groups (AuNP, laser, and laser-activated AuNP groups). Thus, it is possible to conclude that the presence of the AuNPs or laser alone, and the combination of AuNPs + Laser did not result in apparent toxicity. Moreover, histological analysis of the same organs (kidney, spleen, and liver), as well as the administration site, was performed and the results are presented in Figure 7. The animals in the AuNPs-only group (that did not receive laser irradiation) exhibited abundant black granular material (AuNPs) in their subcutaneous and muscle tissues, without any signs of inflammation or toxicity. Furthermore, the animals in the groups that received laser irradiation (laser-only group and AuNPs + Laser group) presented different degrees of necrosis. In the case of the laser-only group, only one animal exhibited necrosis at the irradiated site, which affected the epidermis and dermis; the other animals in this group did not exhibit any necrosis at the irradiated site. Meanwhile, all the animals in the AuNPs + Laser group exhibited necrosis at the irradiated site, which affected the epidermis and dermis, and one exhibited necrosis in the epidermis, dermis, and subcutaneous tissue. Regarding the main excretion organs (spleen, kidney, and liver), no histological findings were noted.

The biodistribution of the AuNPs in these groups was assessed by quantifying the elemental gold in the main excretion organs (liver, spleen, kidney, and blood) and the elemental gold retained at the administration site. The results of the gold quantification by ICP-OES are presented in Figure 8. In the case of the inactivated AuNPs, 82.6% ± 20.0% of the administered gold was retained at the administration site, while 0.1% ± 0.1% was retained in the liver and 0.3% ± 0.1% in the blood. For the laser-activated AuNPs, 87.3% ± 19.9% of the administered gold was retained at the administration site, and 0.2% ± 0.1% was found in the blood. No gold was found in the spleen or in the kidney in both inactivated and laser-activated AuNP groups, and in the case of the laser-activated AuNP group, no gold was found in the liver. This indicates that not only was the majority of the administered AuNPs retained at the administration site 24 h post-administration, but the laser activation did not negatively impact this retention at the administration site.

It is known that the average size of NPs, including AuNPs, significantly impacts their biodistribution profile and accumulation at the tumor site [61,62]. The biodistribution of AuNPs is size-dependent and they have a mean size of around 100 nm, as described by De Jong et al. They demonstrated that 24 h after i.v. administration into the tail vein of rats, ~45% of the administered dose of AuNPs with a size of 100 nm was present in the blood, and ~45% of the administered dose was present in the liver [63]. NPs with a size smaller than 200 nm are less likely to be taken up by the reticuloendothelial system and to accumulate in the liver and spleen [64]. For AuNP-mediated PTT of HNC, our goal is to administer the AuNPs into the tumor and for the AuNPs to accumulate in situ at the administration site and not disseminate to other tissues or organs. Since the mice used to assess the AuNP biodistribution did not exhibit tumors, the AuNPs were administered into the neck region of WT mice, a site supported by the literature, and due to the AuNPs’ mean size, they were retained at the administration site.

### 3.8. Biodistribution and Safety After 30 Days

To study the retention of the AuNPs at the administration site in live animals, female BALB/c mice received a single s.c. injection of the AuNPs (3.1 mg/kg of body weight) into the neck region and subjected to CT at different time points (0, 15, and 30 days post-administration). Figure 9 presents representative CT images of representative mice over the 30-day span, showcasing that the AuNPs remained at the local administration site for up to 30 days post-administration. Moreover, during the 30 days, none of the animals presented any signs of suffering, reinforcing the safety of the inactivated AuNPs.

To further verify that at 30 days post-administration, the majority of the injected AuNPs remained at the administration site, the animals were sacrificed and the main organs of excretion (liver, blood, spleen, and kidney), as well as the administration site, similar to the previous assessment, were collected and analyzed by ICP-OES in order to quantify the amount of elemental gold present. The results are presented in Figure 10. It was found that 80.0% ± 8.4% of the administered gold was retained at the administration site 30 days post-injection, solidifying the idea that the produced AuNPs can stay localized to the region of interest (i.e., administration site). Moreover, no gold was detected in any of the analyzed organs (liver, blood, spleen, or kidneys). Altogether, these results suggest that the developed AuNPs are safe in the short term as they are retained at the administration site. However, more studies with longer follow-up periods are required to ensure that the formulation is also safe in the long term.

## 4. Conclusions

This study investigated the potential application of a novel AuNP formulation for AuNP-mediated PTT to treat HNC, the sixth most prevalent malignancy worldwide. The formulation was thoroughly characterized regarding its physicochemical properties, sterility, and stability over a 3-month storage period. Comprehensive safety assessments demonstrated that the AuNPs were non-toxic in vitro, in CAMs, and in vivo using murine models.

The therapeutic efficacy of AuNP-mediated PTT against HNC was confirmed in vitro across different histological subtypes of the disease. In addition, the in vivo biodistribution studies revealed that the nanoparticles remained at the administration site for at least 30 days post-injection, with no signs of systemic dissemination or associated toxicological effects during this period. Taken together, these findings suggest that the developed AuNP formulation is stable, biocompatible, and capable of achieving sustained local retention—features that are critical for effective PTT. Although further preclinical and clinical studies are needed to optimize the treatment parameters, assess long-term safety, and validate therapeutic efficacy in larger cohorts, the present results support the potential of this AuNP-mediated PTT approach as a promising therapeutic strategy that warrants further investigation for HNC.

From a clinical perspective, the demonstrated safety and biodistribution profiles highlight the potential of this formulation as a minimally invasive and well-tolerated adjuvant to current therapeutic regimens for HNC. By enabling site-specific photothermal ablation with limited systemic exposure, this strategy could contribute to reducing treatment-related toxicity while overcoming the resistance to conventional therapies, as stated in the introduction. At present, most nano-based therapies for HNC under active clinical investigation are focused on optimizing the delivery and efficacy of existing chemotherapeutic agents, which differs from the photothermal approach explored herein. However, despite encouraging progress, key limitations remain, including variability in tumor uptake, challenges in large-scale GMP production, and the lack of standardized treatment protocols. These gaps underscore the need for alternative strategies, such as AuNP-mediated PTT, that can achieve localized therapeutic effects while minimizing systemic exposure. The next steps toward clinical translation will involve optimization of the dosing and irradiation parameters and long-term safety and immunogenicity studies to pave the way for the integration of AuNP-mediated PTT into the multidisciplinary management of HNC.

## Figures and Tables

**Figure 1 pharmaceutics-17-01283-f001:**
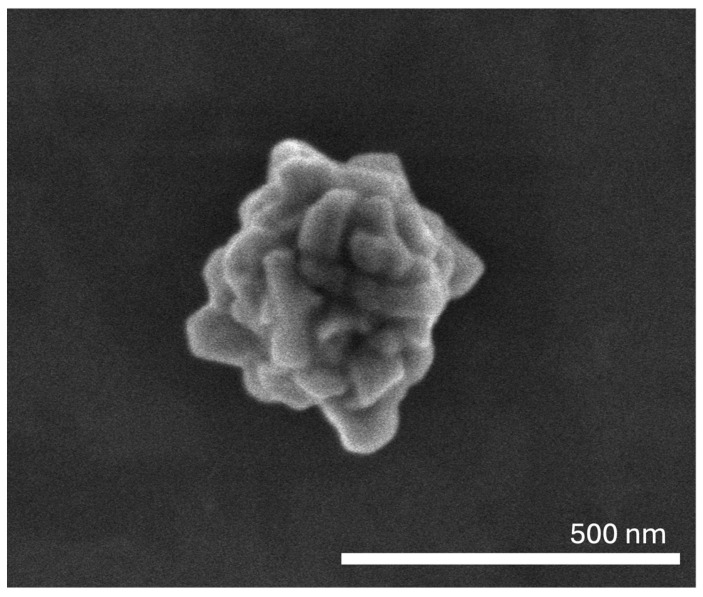
SEM image of the prepared AuNPs.

**Figure 2 pharmaceutics-17-01283-f002:**
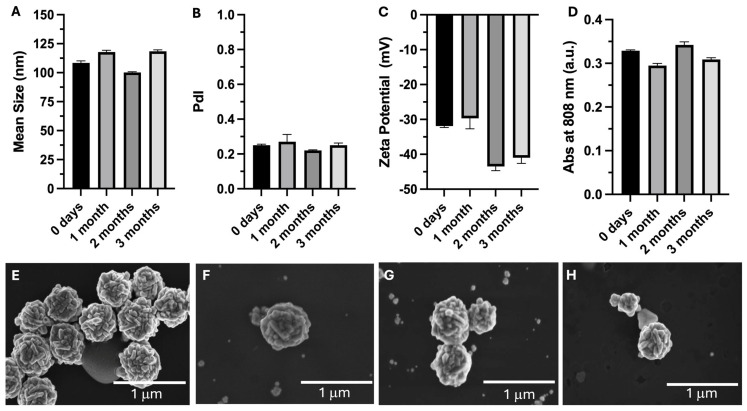
Mean size (**A**), polydispersity index (PdI) (**B**), surface charge (zeta potential) (**C**), absorbance at 808 nm (Abs at 808 nm) (**D**), and morphology of AuNPs at different time points (0 days (**E**), 1 month (**F**), 2 months (**G**), and 3 months (**H**)). AuNPs were kept in water and stored at 4 °C.

**Figure 3 pharmaceutics-17-01283-f003:**
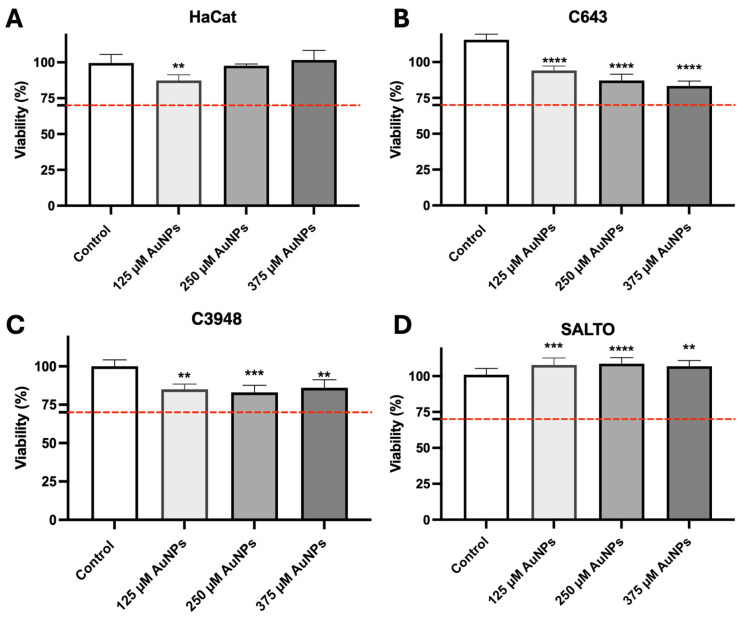
Cell viability of HaCat (**A**) (human keratinocytes), C643 (**B**) (human ATC), C3948 (**C**) (human ATC), and SALTO (**D**) (murine HNSCC) cell lines after a 24 h incubation period with the AuNPs at three different concentrations of gold: untreated (white columns); 125 µM of gold (light grey columns); 250 µM of gold (intermediate grey columns); and 375 µM of gold (dark grey columns). The dashed red line corresponds to 70% cell viability. The results are presented as the mean ± SD; *n* ≥ 6 (** *p* < 0.0021, *** *p* < 0.0002, **** *p* < 0.0001 vs. untreated cells (Control)).

**Figure 4 pharmaceutics-17-01283-f004:**
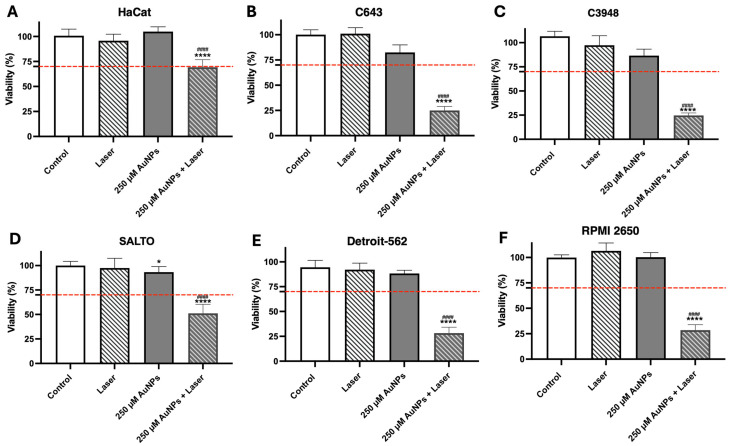
Cell viability of HaCat (**A**) (human keratinocytes), C643 (**B**) (human ATC), C3948 (**C**) (human ATC), SALTO (**D**) (murine HNSCC), Detroit-562 (**E**) (human HNSCC), and RPMI 2650 (**F**) (human HNSCC) cell lines after a 4 h incubation period with the AuNPs at a concentration of 250 µM of gold, with (solid columns) and without laser activation (5.75 W/cm^2^, 5 min) (striped columns). The dashed red line corresponds to 70% cell viability. The results are presented as the mean ± SD; *n* ≥ 6, (* *p* < 0.0332, **** *p* < 0.0001 vs. untreated cells (Control); ^####^ *p* < 0.0001 vs. inactive AuNPs (250 µM of gold)).

**Figure 5 pharmaceutics-17-01283-f005:**
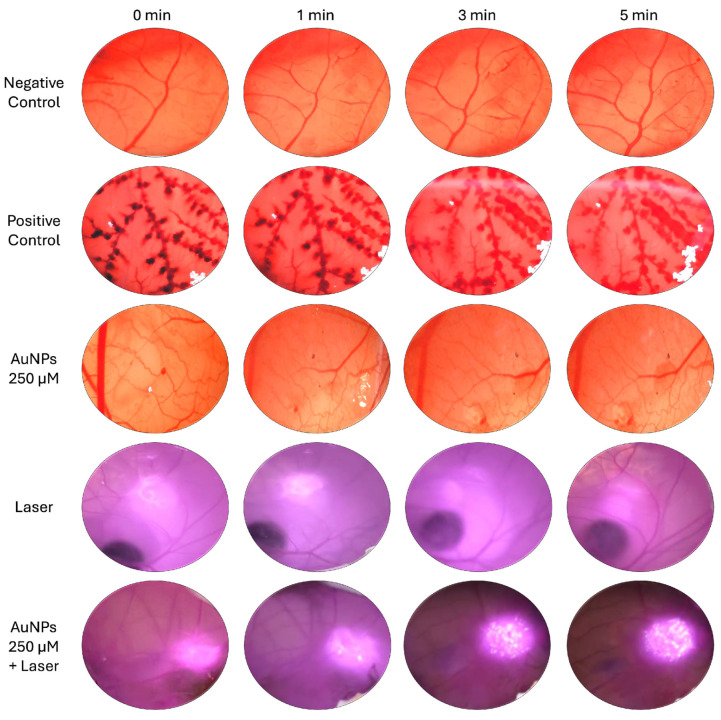
Hen’s Egg Test on the Chorioallantoic Membrane (HET-CAM) exposed to AuNPs (250 µM of gold), laser irradiation (5.75 W/cm^2^), and AuNPs (250 µM of gold) activated by laser irradiation (5.75 W/cm^2^) at different time points (0, 1, 3, and 5 min). Negative (C−) and positive (C+) controls were PBS (pH 7.4) and NaOH (0.1 M), respectively.

**Figure 6 pharmaceutics-17-01283-f006:**
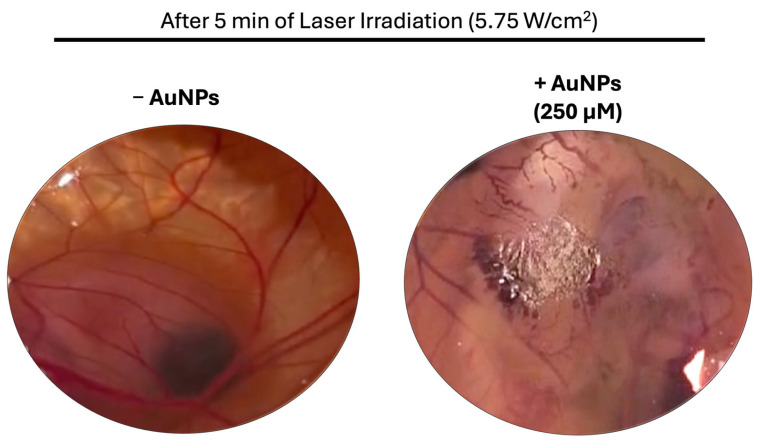
Representative images of CAM after 5 min of laser irradiation (5.75 W/cm^2^) (−AuNPs) or 5 min of exposure to laser (5.75 W/cm^2^)-irradiated AuNPs (250 µM of gold) (+AuNPs) showing visible burn caused by laser-irradiated AuNPs.

**Figure 7 pharmaceutics-17-01283-f007:**
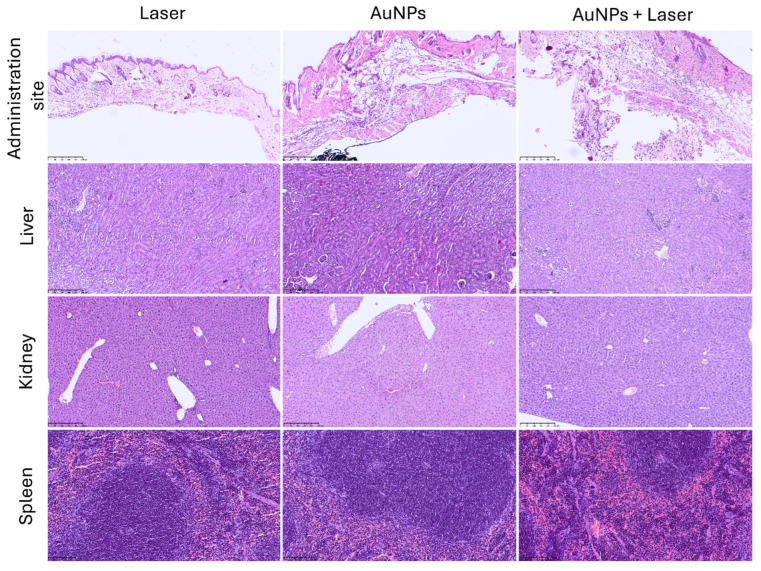
Histological images of the administration site, liver, kidney, and spleen (administration site, liver, and kidney: 100× magnification; spleen: 200× magnification) of animals treated with AuNPs, laser, or laser-activated AuNPs (AuNPs + Laser) (H&E staining).

**Figure 8 pharmaceutics-17-01283-f008:**
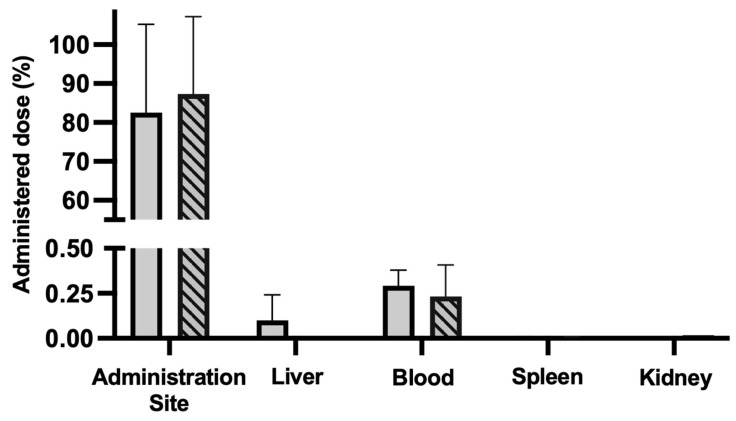
Gold content (ICP-OES) in the administration site, liver, blood, spleen, and kidney of mice 24 h post-s.c. administration of AuNPs (3.1 mg/kg of body weight; solid bars) and laser-activated AuNPs (5.75 W/cm^2^, 5 min; striped bars) as a percentage of the administered dose. The results are presented as the mean ± SD; *n* ≥ 3.

**Figure 9 pharmaceutics-17-01283-f009:**
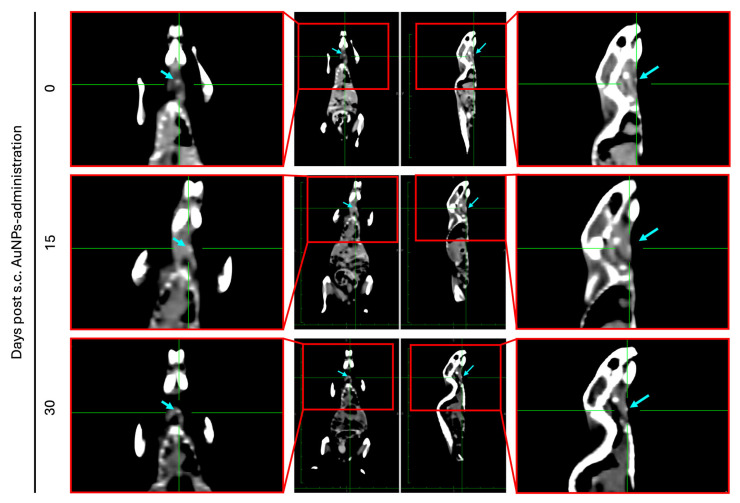
Representative in vivo CT images of female BALB/c mice showing local accumulation of AuNPs (blue arrow) at the site where the AuNPs (3.1 mg/kg of body weight) were administrated s.c. Coronal (left images) and sagittal (right images) planes of a representative animal at different time points (0, 15, and 30 days post-administration) are shown. Green lines represent the axes, and the red squares mark the regions of the images that were magnified (shown as peripheral images).

**Figure 10 pharmaceutics-17-01283-f010:**
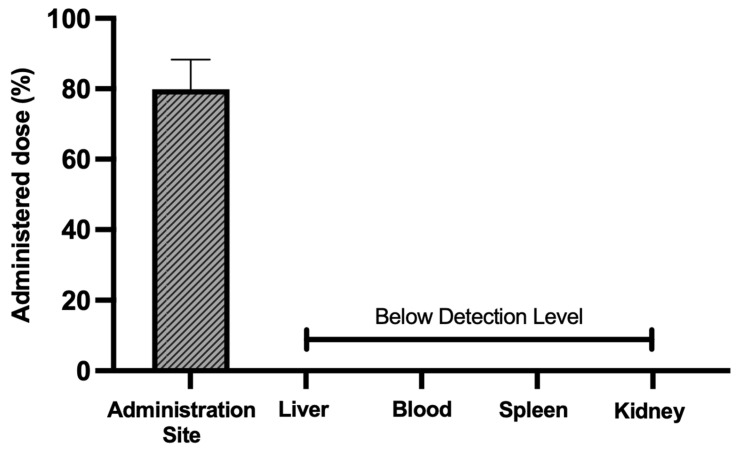
Gold content (ICP-OES) in the administration site, liver, blood, spleen, and kidney of mice 30 days post s.c. administration of AuNPs (3.1 mg/kg of body weight) as percentage of administered dose. The results are presented as the mean ± SD, *n* ≥ 3.

**Table 1 pharmaceutics-17-01283-t001:** AuNPs’ mean size (nm), polydispersity index (PdI), surface charge (zeta potential, mV), and absorbance at 808 nm. The AuNPs were tested at a gold concentration of 250 µM.

	Mean Size (nm)	PdI	Zeta Potential (mV)	Abs_808nm_ (a.u.)
AuNPs	118.2 ± 7.1	0.27 ± 0.05	−32.9 ± 0.4	0.342 ± 0.007

**Table 2 pharmaceutics-17-01283-t002:** Tissue indexes (kidneys, spleen, and liver) of mice 24 h post-treatment: laser irradiation (5.75 W/cm^2^, 5 min); s.c. administration of AuNPs (3.1 mg/kg of body weight); and s.c. administration of AuNPs (3.1 mg/kg of body weight) followed by laser irradiation (5.75 W/cm^2^, 5 min).

Tissue Indexes (Mean ± SD)			
	Kidney	Spleen	Liver
**Laser**	12.7 ± 0.4	6.1 ± 0.1	23.4 ± 0.5
**AuNPs**	12.4 ± 0.6	6.4 ± 0.2	22.6 ± 0.7
**AuNPs + Laser**	12.9 ± 0.5	6.6 ± 0.3	22.5 ± 3.0

## Data Availability

The raw data supporting the conclusions of this article will be made available by the authors on request.

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
