# Peer review of "Towards a Less Invasive Treatment for Head and Neck Cancer: Initial Evaluation of Gold Nanoparticle-Mediated Photothermal Therapy"

_pharmaceutics, 2025, doi:10.3390/pharmaceutics17101283_

Round 1

Reviewer 1 Report

Comments and Suggestions for Authors

This manuscript presents a preliminary study of gold nanoparticle-mediated photothermal therapy (PTT) for head and neck cancer, evaluating the efficiency, biocompatibility, colloidal stability, and in vivo accumulation of the nanoparticles. There are major aspects that need to be addressed:

1) In the title, the phrase “Preclinical Assessment” should be modified (e.g., to “Initial Evaluation”), as the manuscript lacks a comprehensive preclinical evaluation, including full pharmacokinetic, biodistribution, efficacy, and toxicological studies across different animal models.

2) The written part of the manuscript should be revised carefully, as there are some typographical errors. Here are five examples of such errors:

Line 52: “All-together” is incorrect and should be replaced with Altogether.

Lines 70–73 and 74–77 are repetitions and should be merged or shortened.

Line 94: [14,15,24–29,16–23] should be replaced with [14,15,16–23,24–29].

Line 118: HAuCl4.3H2O should be typed as “HAuCl4·3H2O” (use middle dot symbol (·).

Line 118-128: L-ascorbic acid, silver nitrate, and rosmarinic acid should be written in lowercase because they are in the middle of a sentence.

3) In paragraph 4 of the Introduction section, it would be better to compare gold nanoparticles with other nanostructures used in photothermal therapy and support this comparison with updated references.

4) There is an excessive number of self-citations by the authors (nearly 40%).

5) In Section 2, "Materials and Methods", it would be better to add an "Instruments" subsection and move all instrumental details there to improve the fluency of the text.

6) In Section 2.2, some cell lines were provided by individuals. Instead of using their names in the main text, it is better to mention the university or institute from which the cells were obtained, and include the individuals' names in the Acknowledgements section.

7) The titles and subtitles should be uniform in terms of font, size, and formatting.

8) In Section 2.4.2, the authors mention: "For the first two characterizations, AuNPs were diluted in Milli-Q water (1:10, v/v, AuNPs:Milli-Q water) and in PBS (1:10, v/v, AuNPs:PBS) to determine the zeta potential." This raises the question of how surface size can be accurately measured in a high ionic strength medium like PBS. Therefore, this section should be revised for clarity and accuracy.

9) For the morphology assessment using SEM, it would be helpful to mention the concentration of the AuNP solution used for the measurement.

10) In the "3. Results and Discussion" section, it would be helpful to explain the mechanism of colloidal stability of the AuNPs based on their surface chemistry.

11) In Table 1, the value "-32.9 ± 0.4" should be changed to "–32.9 ± 0.4" because the correct minus symbol is the en dash (–).

12) In Figure 1, the AuNPs appear to be composed of small rod-like particles. It would be better to explain this morphology and, more importantly, how the particle shape influences the photothermal properties.

13) In Figure 2C, it would be helpful to explain the ≈10% change in surface negative charge after 2 months (and 3 months), compared to the initial value.

14) In Section 3.8, it would be useful to explain how the AuNPs are eventually cleared from the body, given the long-term toxicological concerns associated with nanoparticles that remain in the body for extended periods.

15) Because of the importance of this research, it is recommended to add a final section on the current preclinical (or clinical) stages of nano-based therapies for head and neck cancers, and to include current limitations and research gaps.

16) The quality of Figures 3, 4, and 8–10 should be improved.

17) It is recommended to double-check the reference section. For example, in References 7, 11, and 22, the word '(Basel).' should be removed from the journal name.

Comments on the Quality of English Language

The quality of the English language should be improved, as there are some typographical errors. The titles and subtitles should be uniform in terms of font, size, and formatting.

Author Response

Reviewer 1 -

This manuscript presents a preliminary study of gold nanoparticle-mediated photothermal therapy (PTT) for head and neck cancer, evaluating the efficiency, biocompatibility, colloidal stability, and in vivo accumulation of the nanoparticles. There are major aspects that need to be addressed:

1) In the title, the phrase “Preclinical Assessment” should be modified (e.g., to “Initial Evaluation”), as the manuscript lacks a comprehensive preclinical evaluation, including full pharmacokinetic, biodistribution, efficacy, and toxicological studies across different animal models.

R1: We thank the Reviewer for this valuable suggestion. We agree that the original wording could be misleading, as the current study does not encompass the full scope of a comprehensive preclinical assessment. Accordingly, we have revised the title to more accurately reflect the preliminary nature of our work. The new title is:

“Towards a Less Invasive Treatment for Head and Neck Cancer: Initial Evaluation of Gold Nanoparticle-Mediated Photothermal Therapy.”

2) The written part of the manuscript should be revised carefully, as there are some typographical errors. Here are five examples of such errors:

Line 52: “All-together” is incorrect and should be replaced with Altogether.

Lines 70–73 and 74–77 are repetitions and should be merged or shortened.

Line 94: [14,15,24–29,16–23] should be replaced with [14,15,16–23,24–29].

Line 118: HAuCl4.3H2O should be typed as “HAuCl4·3H2O” (use middle dot symbol (·).

Line 118-128: L-ascorbic acid, silver nitrate, and rosmarinic acid should be written in lowercase because they are in the middle of a sentence.

R2: Thank you for the suggestions. The mentioned typographical errors were corrected, and the manuscript was carefully revised.

3) In paragraph 4 of the Introduction section, it would be better to compare gold nanoparticles with other nanostructures used in photothermal therapy and support this comparison with updated references.

R3: Thank you for the suggestion, the reviewer’s concern has been addressed, and gold nanoparticles were compared to other nano-based PTAs, and the reasons why AuNPs are advantageous were mentioned, as follows:

“(…) Besides AuNPs, there are many nano-based PTA’s made from other materials such as copper selenide [15], carbon [16,17], tungsten [18,19], graphene oxide [16], black phosphorus [20], copper [21,22], etc [23]. However, although these nonsystems present very promising optical properties for PTA applications, they face several limitations in terms of low biocompatibility that can lead to possible toxicity [24–28]. Moreover, PTA’s based on the mentioned materials are usually synthetized through complex processes and the resulting nano-PTA’s present poor colloidal stability and undergo oxidative degradation [26,27,29]. On the other hand, AuNPs are chemically inert, biocompatible, and present enhanced photothermal conversion ability due to the Localized Surface Plasmon Resonance Effect (LSPR). (…)”

4) There is an excessive number of self-citations by the authors (nearly 40%).

R4: We thank the Reviewer for this observation. We carefully re-evaluated the reference list. While many of our previous works were directly relevant and contributed to the discussion, we have revised the manuscript accordingly. We have reduced the number of self-citations and complemented the discussion with additional, independent references to ensure a more balanced and comprehensive contextualization of the study.

5) In Section 2, "Materials and Methods", it would be better to add an "Instruments" subsection and move all instrumental details there to improve the fluency of the text.

R5: Thank you for the suggestion. Although the “Intruments” section is not present in Pharmaceutics’ “Instructions for Authors” it was added to the manuscript in order to meet the Reviewers’ request:

“Heidolph magnetic stirrer hotplate MR3001 (Heidolph Instruments, Schwabach, Germany), Z 327 K centrifuge (Hermle LaborTechnik GmbH, Wehingen, Germany), Ultima Spectrometer (Horiba Jobin-Yvon, Longjumeau, France), Zetasizer Nano S (Malvern Instruments, Malvern, UK), Zetasizer Nano Z (Malvern Instruments, Malvern, UK), Hitachi Regulus 8220 Scanning Electron Microscope (Hitachi, Mito, Japan), Shimadzu UV-1280 (Shimadzu, Corp., Kyoto, Japan), Microfil™ filtration system (Merck Millipore, Burlington, MA, USA), BioTekTM EL x800TM Absorbance Microplate Reader (Fisher Scientific, NH, USA), Olympus CX51 microscope (Olympus Corporation, Tokyo, Japan), NanoZoomer-SQ Digital slice scanner – C13140-01 (Hamamtsu, Japan), Toshiba Astelion 16 CT scanner (Toshiba, Tokyo, Japan).”

6) In Section 2.2, some cell lines were provided by individuals. Instead of using their names in the main text, it is better to mention the university or institute from which the cells were obtained, and include the individuals' names in the Acknowledgements section.

R6: Thank you for this suggestion. We have revised Section 2.2 so that it now only contains information regarding the maintenance of the cell lines. The details related to the donation of the cells, including the names of the individuals and their respective universities/institutes, have been moved to the Acknowledgments section.

7) The titles and subtitles should be uniform in terms of font, size, and formatting.

R7: Thanks to the Reviwer for the suggestion. The titles and subtitles were uniformized.

8) In Section 2.4.2, the authors mention: "For the first two characterizations, AuNPs were diluted in Milli-Q water (1:10, v/v, AuNPs:Milli-Q water) and in PBS (1:10, v/v, AuNPs:PBS) to determine the zeta potential." This raises the question of how surface size can be accurately measured in a high ionic strength medium like PBS. Therefore, this section should be revised for clarity and accuracy.

R8: Thank you to the Reviewer for the question. The section was revised, as follows, for clarity and accuracy. However, we would like to clarify to the Reviewer that size was not measured in a high ionic stregth medium (PBS), but in water. In contrast, surface charge was measured in a buffer medium, PBS, as is required for the technique used. The choice of PBS as a medium to determine surface charge is due to this being the buffer used to administrate the AuNPs in the in vivo assessments.

“Mean size, polydispersity index (PdI) and surface charge of the AuNPs was determined. AuNPs were diluted in Milli-Q water (1:10, v/v, AuNPs:Milli-Q water) to determine mean size and polydispersity index (PdI) by Dynamic Light Scattering (DLS), and in PBS (1:10, v/v, AuNPs:PBS) to determine the zeta potential by Electrophoretic Mobility Assay. All measurements were performed ≥ 3 times. ”

9) For the morphology assessment using SEM, it would be helpful to mention the concentration of the AuNP solution used for the measurement.

R9: Thank you for the suggestion. The concentration of the solution of AuNPs used in SEM has been added.

10) In the "3. Results and Discussion" section, it would be helpful to explain the mechanism of colloidal stability of the AuNPs based on their surface chemistry.

R10: The developed AuNPs present colloidal stability through electrostatic stabilization, and this information was added to the respective section (3.1.), as follows:

“Negatively charged AuNPs with zeta potential values below -30 mV are considered physically stable due to sufficient electrostatic repulsion, preventing nanoparticle aggregation [50,51]. In agreement with this, the AuNPs developed in this study exhibit negative surface charges. Moreover, and as demonstrated in previous work, the developed AuNPs only present gold in their composition, thus evidencing a non-reactive surface resultant from the inert nature of this metal [31]. Together, these features are expected to provide long-term colloidal stability through electrostatic stabilization and the intrinsic chemical inertness of gold.”

11) In Table 1, the value "-32.9 ± 0.4" should be changed to "–32.9 ± 0.4" because the correct minus symbol is the en dash (–).

R11: Thank you to the Reviewer for pointing out this typo. The minus sign was corrected to the en dash (–) accordingly.

12) In Figure 1, the AuNPs appear to be composed of small rod-like particles. It would be better to explain this morphology and, more importantly, how the particle shape influences the photothermal properties.

R12: More details on the morphology of the AuNPs shown in Figure 1 was added. This morphology has been previously described as a nanoflower-like morphology derived from nanosheet aggregations, which enhance photothermal conversion due to the presence of “hot spots.” The following was added:

“This morphology seems to originate from the controlled agglomeration of gold nanosheets [31,52,53]. Nanoflowers present enhanced photothermal conversion due to the presence of branches, as these structures act as “hot spots” [53,54].”

13) In Figure 2C, it would be helpful to explain the ≈10% change in surface negative charge after 2 months (and 3 months), compared to the initial value.

R13: Thank you for this observation. The zeta potential shifted from −32.9 mV at t = 0 to −43 mV at 2 months and −41 mV at 3 months. As the AuNPs are uncoated and stored in Milli-Q water, this change is most likely due to gradual surface adsorption of hydroxyl and trace anionic species from aqueous medium, which can alter the electrostatic double layer at the gold–water interface. Importantly, mean size and PdI remained stable, confirming that the colloidal suspension remained stable despite the observed variation in surface potential. This explanation has been added to the Results/Discussion section.

14) In Section 3.8, it would be useful to explain how the AuNPs are eventually cleared from the body, given the long-term toxicological concerns associated with nanoparticles that remain in the body for extended periods.

R14: Thank you to the Reviewer for raising this point. We have demonstrated in this work, and in other work from the group with a similar formulation (Ferreira-Gonçalves, T. et al., Int. J. Pharm. 2024), that the AuNPs are retained at the administration site for a long period of time (at least 30 days). Moreover, in previous work with a similar formulation, the group has demonstrated that even at much higher concentrations, the AuNPs are still safe (Lopes, J. et al., Pharmaceutics 2023).

Given their relatively large size, the developed AuNPs are above the renal clearance threshold (>10 nm, Soo Choi, H. et al., Nat Biotechnol. 2007; Longmire H. et al., Nanomedicine (Lond). 2012) and also above the typical cutoff for hepatic clearance (> 20 nm, Longmire H. et al., Nanomedicine (Lond). 2012).

Therefore, elimination through these physiological pathways is unlikely. Our working hypothesis, and intended application of this formulation, is that the AuNPs will be removed along with the necrosed tumor tissue during surgical excision following AuNP-mediated PTT. 

15) Because of the importance of this research, it is recommended to add a final section on the current preclinical (or clinical) stages of nano-based therapies for head and neck cancers, and to include current limitations and research gaps.

R15: Thanks to the Reviewer for the valuable recommendation. Following the suggestion, we added a final paragraph to the Conclusion section, discussing the current preclinical/clinical landscape of nano-based therapies for HNC, highlighting limitations such as variability in tumor uptake, GMP production challenges, and lack of standardized protocols, while positioning AuNP-mediated PTT as a promising alternative, as follows:

“From a clinical perspective, the demonstrated safety and localized biodistribution profiles highlight the potential of this formulation as a minimally invasive and well-tolerated adjuvant to current therapeutic regimens for HNC. By enabling site-specific photothermal ablation with limited systemic exposure, this strategy may contribute to reduce treatment-related toxicity and overcoming resistance to conventional therapies, as stated in the introduction. At present, most nano-based therapies for HNC under active clinical investigation are focused on optimizing the delivery and efficacy of existing chemotherapeutic agents which differ from the photothermal approach herein explored. However, despite encouraging progress, key limitations remain, including variability in tumor uptake, challenges in large-scale GMP production, and the lack of standardized treatment protocols. These gaps underscore the need for alternative strategies, such as AuNP-mediated PTT, that can achieve localized therapeutic effects while minimizing systemic exposure. The next steps toward clinical translation will involve optimization of dosing and irradiation parameters, long-term safety and immunogenicity studies, paving the way for the integration of AuNP-mediated PTT into the multidisciplinary management of HNC.”

16) The quality of Figures 3, 4, and 8–10 should be improved.

R16: Thank you for the suggestion. The Figures 3,4,8-10 were replaced with higher-quality Figures.

17) It is recommended to double-check the reference section. For example, in References 7, 11, and 22, the word '(Basel).' should be removed from the journal name.

R17: Thank you for the suggestion. The references section were double-checked and this has been corrected.

Reviewer 2 Report

Comments and Suggestions for Authors

This manuscript reports the synthesis, characterization, and preclinical evaluation of gold nanoparticles (AuNPs) for photothermal therapy (PTT) in head and neck cancer (HNC). The authors demonstrate that the AuNPs are stable, sterile, and biocompatible in vitro and in ovo, while exhibiting significant cancer cell killing upon NIR laser activation. Importantly, AuNPs remained locally retained at the injection site for up to 30 days without systemic dissemination or toxicity, highlighting their potential for localized PTT applications. The study is well-structured, provides comprehensive characterization, and applies multiple preclinical models. However, the novelty of this manuscript appears limited. The nanoparticle formulation has already been reported previously by the same group, and the concept of using AuNPs for photothermal therapy in HNC is well established in the literature. Thus, the present work mainly extends prior findings by providing additional preclinical safety and retention data rather than offering a fundamentally new therapeutic strategy. The authors should more clearly articulate what differentiates this study from their earlier publications and from other AuNP-based PTT studies, and justify how this work significantly advances the current state of knowledge.

1. The quality of SEM image in Figure 1 is poor. In the SEM images, it seems the nanoparticles are aggregated.

2. Photothermal studies needed to performed and data should be included. Photothermal conversion efficiency should be calculated.

3. It is suggested to provide graphical abstract for the work.

Author Response

This manuscript reports the synthesis, characterization, and preclinical evaluation of gold nanoparticles (AuNPs) for photothermal therapy (PTT) in head and neck cancer (HNC). The authors demonstrate that the AuNPs are stable, sterile, and biocompatible in vitro and in ovo, while exhibiting significant cancer cell killing upon NIR laser activation. Importantly, AuNPs remained locally retained at the injection site for up to 30 days without systemic dissemination or toxicity, highlighting their potential for localized PTT applications. The study is well-structured, provides comprehensive characterization, and applies multiple preclinical models. However, the novelty of this manuscript appears limited. The nanoparticle formulation has already been reported previously by the same group, and the concept of using AuNPs for photothermal therapy in HNC is well established in the literature. Thus, the present work mainly extends prior findings by providing additional preclinical safety and retention data rather than offering a fundamentally new therapeutic strategy. The authors should more clearly articulate what differentiates this study from their earlier publications and from other AuNP-based PTT studies, and justify how this work significantly advances the current state of knowledge.

  1. The quality of SEM image in Figure 1 is poor. In the SEM images, it seems the nanoparticles are aggregated.

R1: The authours would like to thank the Reviewer for the mentioned. Figure 1 was replaced with a Figure with higher quality. The apparent aggregation of AuNPs observed in SEM images is most likely related to sample preparation rather than to the intrinsic properties of the nanoparticles. During the drying process required for SEM, solvent evaporation can induce particle clustering due to capillary forces. In addition, SEM provides a two-dimensional projection of the dried sample, so particles in close proximity may appear as aggregates even if they are not chemically bound. In contrast, PdI measured by DLS reflects the actual state of the nanoparticles in suspension. The low PdI value obtained confirms that the formulation is homogeneous/monodisperse in size distribution, supporting the conclusion that the aggregation seen in SEM images is an artifact of sample preparation rather than a characteristic of the AuNP formulation.

  1. Photothermal studies needed to performed and data should be included. Photothermal conversion efficiency should be calculated.

R2: Thank you for the suggestion. Photothermal conversion studies were thoroughly performed during the formulation development and optimization stage, and the results were already published (Neves Amaral M., et al., J. Drug Deliv. Sci. Technol. 2024). Since the formulation protocol remained unchanged, the photothermal properties are expected to remain consistent and reproducible, as confirmed in our previous work. In the present manuscript, the main objective was to evaluate the safety and therapeutic efficacy of the optimized formulation in biological systems. Repeating photothermal experiments under identical conditions would therefore be redundant and beyond the scope of this study. To address the Reviewer’s concern, we cited the published data and included the relevant discussion in the revised version of the manuscript, as follows:

“ (…) The prepared AuNPs were shown to also present high molar extinction coefficient at 808 nm (~6 x 108 M-1cm-1) , which translated in high photothermal conversion ability, with AuNPs at 125 µM of gold resulting in a temperature increase of ~10 °C upon laser irradiation (808 nm, 5.6 W/cm2) [31]. (…)”

  1. It is suggested to provide graphical abstract for the work.

R3: Thank you for the suggestion. The following Graphical Abstract was included upon submission of the original manuscript.

Reviewer 3 Report

Comments and Suggestions for Authors

The writing and organization of Amaral et al.'s manuscript, "Towards a Less Invasive Treatment for Head and Neck Cancer: Preclinical Assessment of Gold Nanoparticle-Mediated Photothermal Therapy," are excellent. The authors of this work assess a novel photothermal treatment (PTT) approach for head and neck cancer based on gold nanoparticles (AuNP), showing good in vitro efficacy and biocompatibility across several models. I suggest making a few little changes before publishing in Pharmaceutics.

  1. It is recommended that the authors condense the introduction by minimizing the repetition of details about the difficulties in treating head and neck cancer.
  2. The authors suggested replacing certain figures that might benefit from improved resolution and more lucid labeling, such as SEM pictures and CAM assay results.
  3. The authors emphasize that while the in vivo biodistribution results are encouraging, the safety claims would be strengthened by a longer follow-up period than 30 days.
  4. The authors should highlight its uniqueness, the talk should contrast this study with other recent nanoparticle-based PTT techniques.
  5. The authors want to make it clear if repeated AuNP administrations were considered and how this would impact safety.
  6. The clinical implications and the next steps required for translation into patient care should be more prominently highlighted in the conclusion.
  7. The authors should proofread the article for grammar and typographical errors.

Author Response

The writing and organization of Amaral et al.'s manuscript, "Towards a Less Invasive Treatment for Head and Neck Cancer: Preclinical Assessment of Gold Nanoparticle-Mediated Photothermal Therapy," are excellent. The authors of this work assess a novel photothermal treatment (PTT) approach for head and neck cancer based on gold nanoparticles (AuNP), showing good in vitro efficacy and biocompatibility across several models. I suggest making a few little changes before publishing in Pharmaceutics.

1. It is recommended that the authors condense the introduction by minimizing the repetition of details about the difficulties in treating head and neck cancer.

R1: Thank you for the suggestion. Indeed the difficulties of treating HNC were repeated and were condensed.

2. The authors suggested replacing certain figures that might benefit from improved resolution and more lucid labeling, such as SEM pictures and CAM assay results.

R2: The authors would like to thank the Reviewer for the suggestion. Many Figures of the originally-submitted manuscript were replaced by higher resolution Figures in the Revised manuscript, including the SEM pictures.

3. The authors emphasize that while the in vivo biodistribution results are encouraging, the safety claims would be strengthened by a longer follow-up period than 30 days.

R3: We thank the Reviewer for this insightful comment. We agree that, although the biodistribution and safety profile observed in our study are highly encouraging, the 30-day follow-up period primarily reflects short-term safety of the formulation. It is important to highlight that a 30-day evaluation period is in line with standard practice for preliminary in vivo toxicological and biodistribution assessments of nanoformulations. Several studies investigating metallic nanoparticles, including gold-based systems, have employed similar timelines (typically 14–30 days) to establish early safety profiles and identify potential acute, sub-acute, sub-chronic (Attarde S. S. et al., Curr Pharm Biotechnol 2020) or even chronic (Cardoso E. et al., Mutat Res. 2014) toxicities. Comparable timeframes have also been explicitly described as “chronic” in the literature, for example by Akkam et al. (2023), who investigated the acute and chronic adverse effects of gold nanoparticles over a 28-day period (Akkam N. et al., Drug Dev Ind Pharm. 2023) or by Al-Radadi et al. (2023), who reported Passiflora ligularis-mediated gold nanoparticles “chronically administered from day 8 to day 28” in a 28-day in vivo study (Al-Radadi N.S. et al., Saudi Pharm J. 2023). This timeframe is widely considered sufficient to capture immediate tissue accumulation, potential acute inflammatory responses, and early clearance dynamics from major organs such as the liver, spleen, and kidneys.

Furthermore, our formulation demonstrated a homogeneous size distribution and a stabilizing surface chemistry, both of which are known factors that promote predictable pharmacokinetics and reduce the risk of uncontrolled aggregation or long-term tissue retention.

These physicochemical characteristics, combined with the absence of histopathological alterations or clinical signs of toxicity within 30 days, provide strong evidence supporting the short-term safety of the system.

Nonetheless, we fully acknowledge that long-term monitoring will be an important step to further evaluate potential chronic effects and delayed clearance pathways. As such, we have clarified in the revised manuscript that while our present results confirm short-term safety, future studies with extended follow-up periods will be planned to strengthen claims of long-term safety and translational potential.

4. The authors should highlight its uniqueness, the talk should contrast this study with other recent nanoparticle-based PTT techniques.

R4: Thank you for the suggestion. The advantages of using AuNPs in PTT versus other nanoparticle-based PTT approaches were added to the Introduction section, as follows:

“(…) Besides AuNPs, there are many nano-based PTA’s made from other materials such as copper selenide [15], carbon [16,17], tungsten [18,19], graphene oxide [16], black phosphorus [20], copper [21,22], etc [23]. However, although these nonsystems present very promising optical properties for PTA applications, they face several limitations in terms of low biocompatibility that can lead to possible toxicity [24–28]. Moreover, PTA’s based on the mentioned materials are usually synthetized through complex processes and the resulting nano-PTA’s present poor colloidal stability and undergo oxidative degradation [26,27,29]. (…)”

5. The authors want to make it clear if repeated AuNP administrations were considered and how this would impact safety.

R5: The authors would like to thank the Reviewer for the pertinent question. Repeated AuNP administrations were indeed considered but deemed unnecessary for our therapeutic design. Instead of performing multiple administrations of AuNPs, our strategy relies on multiple irradiations of the same pool of AuNPs after a single administration. In line with prior in vivo protocols, we think that a single intratumoral administration followed by multiple NIR irradiations of the same nanoparticle depot might limit potential cumulative systemic exposure while maintaining efficacy like described in the precedent literature (e.g., Lopes et al., single injection + three irradiations (Lopes J. et al., Int J Pharm. 2025); Domingo-Diez et al., single injection + three consecutive irradiations; and fractionated PTT paradigms (Domingo-Diez J. et al., Int J Nanomedicine. 2025). These reports confirm that the strategy of single administration combined with multiple irradiations is both feasible and advantageous, and align with our strategy. For this reason, it is critical that the AuNPs remain localized at the administration site for a sufficient period of time to allow multiple irradiation sessions. Our biodistribution results confirmed that the formulation remained at the injection site for at least 30 days, which is in line with what has been reported for similar AuNP-based systems (Ferreira-Gonçalves T. et al., Int J Pharm. 2024, Lopes J. et al., Int J Pharm. 2025). This feature supports the rationale for a single administration followed by multiple irradiation cycles, thereby enhancing both safety and therapeutic feasibility.

6. The clinical implications and the next steps required for translation into patient care should be more prominently highlighted in the conclusion.

R6: Thank you for the suggestion. This has been highlighted in the last paragraph of the conclusion, as follows:

“(…) From a clinical perspective, the demonstrated safety and biodistribution profiles highlight the potential of this formulation as a minimally invasive and well-tolerated adjuvant treatment to current therapeutic regimens for HNC. By enabling site-specific photothermal ablation with limited systemic exposure, this strategy may contribute to reduce treatment-related toxicity while overcoming resistance to conventional therapies, as stated in the introduction. Currently, the active and recruiting clinical trials for HNC related to nanotechnology are focused on improving the already-used chemotherapeutic agents, a very different approach to the one presented in this work. The next steps toward clinical translation will involve optimization of dosing and irradiation parameters, long-term safety and immunogenicity studies, and the establishment of GMP-compatible production protocols. Ultimately, more pre-clinical studies will be required to confirm safety, feasibility, and therapeutic benefit in patients, paving the way for the integration of AuNP-mediated PTT into the multidisciplinary management of HNC. (…)”

7. The authors should proofread the article for grammar and typographical errors.

R7: The authors would like to thank the Reviewer for the suggestion and the manuscript was proofread and the detected typographical errors were corrected.

Round 2

Reviewer 1 Report

Comments and Suggestions for Authors

The manuscript has been substantially revised and is now suitable for consideration for publication in its current form.

Reviewer 2 Report

Comments and Suggestions for Authors

The authors improved the quality of the manuscript. so, it can be accepted now for publication.